# Characterizing large-scale quantum computers via cycle benchmarking

Alexander Erhard[1,7], Joel J. Wallman[2,3,7]*, Lukas Postler[1], Michael Meth[1], Roman Stricker[1],
Esteban A. Martinez[1,4], Philipp Schindler[1], Thomas Monz[1,5]*, Joseph Emerson[2,3] & Rainer Blatt[1,6]

Quantum computers promise to solve certain problems more efficiently than their digital counterparts. A major challenge towards practically useful quantum computing is characterizing and reducing the various errors that accumulate during an algorithm running on large-scale processors. Current characterization techniques are unable to adequately account for the exponentially large set of potential errors, including cross-talk and other correlated noise sources. Here we develop cycle benchmarking, a rigorous and practically scalable protocol for characterizing local and global errors across multi-qubit quantum processors. We experimentally demonstrate its practicality by quantifying such errors in non-entangling and entangling operations on an ion-trap quantum computer with up to 10 qubits, and total process fidelities for multi-qubit entangling gates ranging from $99.6(1)\%$ for 2 qubits to $86(2)\%$ for 10 qubits. Furthermore, cycle benchmarking data validates that the error rate per single-qubit gate and per two-qubit coupling does not increase with increasing system size.

[1] Institute for Experimental Physics, University of Innsbruck, 6020 Innsbruck, Austria. [2] Institute for Quantum Computing and Department of Applied Mathematics, University of Waterloo, Waterloo, Canada. [3] Quantum Benchmark Inc., Kitchener, ON N2H 4C3, Canada. [4] Niels Bohr Institute, University of Copenhagen, 2100 Copenhagen, Denmark. [5] Alpine Quantum Technologies GmbH, 6020 Innsbruck, Austria. [6] Institute for Quantum Optics and Quantum Information of the Austrian Academy of Sciences, 6020 Innsbruck, Austria. [7] These authors contributed equally: Alexander Erhard, Joel J. Wallman. *email: jwallman@uwaterloo.ca; thomas.monz@uibk.ac.at

 **1**

Practical methods to characterize quantum processes acting on large-scale quantum systems are required to assess current devices and steer the development of future, more powerful devices. In principle, quantum processes can be fully characterized using, for example, quantum process tomography[1] or gate set tomography[2–4]. However, any protocol for fully characterizing a quantum process requires a number of experiments and digital post-processing resources that grows exponentially with the number of qubits, even with improvements such as compressed sensing[5,6]. As a result, the largest quantum processes that have been fully characterized to date acted only on three qubits[7].

The exponential resources required for a full characterization can be circumvented by extracting partial information about quantum processes. A partial characterization typically yields some figure of merit comparing the noisy implementation of a quantum process to the desired operation. We will consider the process fidelity (also known as the entanglement fidelity), which is equivalent to the average gate fidelity up to a dimensional factor that is approximately 1[8,9].

The process fidelity can be efficiently estimated by randomized benchmarking[10–12] or direct fidelity estimation[13–15]. Direct fidelity estimation can be efficient and hence has been implemented for up to 7 qubits[16] but conflates state preparation and measurement (SPAM) errors with the process fidelity, limiting its value for realistic systems. SPAM errors increase with the system size and so robustness to SPAM is increasingly important for many qubits. Randomized benchmarking decouples the SPAM errors from gate operation errors by applying multiple random elements of the $N$-qubit Clifford group[11,12]. However, implementing each Clifford operation requires $\mathcal{O}(N^2/\mathrm{log}eN)$ primitive two-qubit operations[17], so that randomized benchmarking provides very coarse information about the primitive operations. Furthermore, for error rates as low as $0.1\%$ per two-qubit operation, a single 10-qubit Clifford operation will have a cumulative error rate on the order of $10\%$, which substantially increases the number of measurements required to accurately estimate the process fidelity.

Owing to these practical limitations, randomized benchmarking has only been applied on operations involving three or less qubits[18]. While randomized benchmarking can be performed on small subsets of the qubit register[19], such experiments do not explore the full Hilbert space and therefore will not detect important performance-limiting error mechanisms such as crosstalk. Moreover, errors in operations must be characterized in the context in which they are used because control sequences for a specific gate are often distorted by other gates performed in parallel. One method to achieve this is to only perform gates in fixed modes of parallel operation. We refer to a parallel set of gates as a cycle, in analogy with a digital clock cycle. In typical architectures, there are two types of cycles, namely, cycles of single-qubit gates and cycles of multi-qubit gates. Undetected calibration and cross-talk errors will typically lead to coherent and spatially correlated errors that can lead to substantially larger algorithmic errors and can require higher overheads in fault-tolerant quantum error correction schemes[20]. Such errors can be converted to stochastic Pauli errors by randomizing the cycles of single-qubit gates in such a way that the overall ideal circuit remains unchanged, a technique known as randomized compiling (RC)[21]. The error rate due to the resulting stochastic Pauli errors can then be accurately quantified by the process fidelity.

In this paper, we introduce cycle benchmarking (CB), a protocol for estimating the process fidelity of a global noise process affecting a quantum device that occur when a cycle of operations is applied to a quantum register. Under the assumption of Markovian noise such that the noise on each cycle of independent single-qubit gates is independent of the specific gates being implemented (see Supplementary Note 1), we prove that CB is robust to SPAM errors and that the number of measurements required to estimate the process fidelity to a fixed precision is approximately independent of the number of qubits. We demonstrate the practicality of CB for many-qubit systems by using it to experimentally estimate the process fidelity of both non-entangling Pauli operations and the multi-qubit entangling Mølmer–Sørensen (MS) gate[22,23] acting on up to ten qubits. We also confirm that the protocol and analysis methods, derived under theoretical assumptions, produce consistent results in our experimental system.

## Results

**The CB protocol.** We now outline how the CB protocol can quantify the effect of global and local error mechanisms affecting different primitive cycle operations of interest.

Mathematically, the ideal operation of interest is described by the corresponding unitary matrix $G$. Its action is expressed by a map $\mathcal{G} : \rho \rightarrow G\rho G^{\dagger}$ that acts on the state of the quantum register, described by the density matrix $\rho$. We denote the map of an ideal operation by capital calligraphic letters, such as $\mathcal{G}$, and their noisy experimental implementations will be indicated by an overset tilde, such as $\tilde{\mathcal{G}}$. We denote the composition of gates by the natural matrix operations for the map representation, so, e.g., $\mathcal{R}\mathcal{G}$ means first apply $\mathcal{G}$ then apply $\mathcal{R}$, and $\mathcal{G}^m$ means apply $\mathcal{G}$ a total of $m$ times. A particularly important class of processes are Pauli cycles $\mathcal{P}$, where the unitary matrix of the process is the $N$-qubit Pauli matrix $P$.

We evaluate the quality of a noisy process $\tilde{\mathcal{G}}$ by its process fidelity to the ideal target $\mathcal{G}$, which can be written as[13]

$$F(\tilde{\mathcal{G}}, \mathcal{G}) = \sum_{P \in \{I,X,Y,Z\}^{\otimes N}} 4^{-N} F_P(\tilde{\mathcal{G}}, \mathcal{G}), \tag{1}$$

where

$$F_P(\tilde{\mathcal{G}}, \mathcal{G}) = 2^{-N} \mathrm{Tr}\big[\mathcal{G}(P)\tilde{\mathcal{G}}(P)\big]. \tag{2}$$

Each quantity $F_P(\tilde{\mathcal{G}}, \mathcal{G})$ can be experimentally estimated by preparing an eigenstate of $P$, applying the noisy gate $\tilde{\mathcal{G}}$, and then measuring the expectation value of the ideal outcome $\mathcal{G}(P)$. The process fidelity may be estimated by averaging $F_P(\tilde{\mathcal{G}}, \mathcal{G})$ over a set of Pauli matrices. However, a sampling protocol (as in direct fidelity estimation[13,14]) for estimating these individual terms is not robust to SPAM errors. Robustness to SPAM is particularly important because SPAM errors can dominate the gate errors.

Inspired by randomized benchmarking[10], SPAM errors can be decoupled from the process fidelity by applying the noisy operation of interest $\tilde{\mathcal{G}}$ a total of $m$ times and extracting process fidelity from the decay of $F_P(\tilde{\mathcal{G}}^m, \mathcal{G}^m)$ as a function of the sequence length $m$. Extracting a meaningful error per application of the gate of interest is nontrivial for generic noise channels[24]. However, decay rates can be extracted straightforwardly for Pauli noise channels, that is, classical mixtures of Pauli operations that are applied to the register randomly with given probability. Mathematically, a Pauli noise channel is a map

$$\mathcal{E} : \rho \rightarrow \sum_{P \in \{I,X,Y,Z\}^{\otimes N}} \mu(P)P\rho P^{\dagger} \tag{3}$$

for some probability distribution $\mu$. Such channels cannot exactly describe, for example, small over-rotation errors or amplitude damping channels.

Since the noise in our system is generic, we want to engineer the noise such that it can be described well by a Pauli noise

channel. It has been shown that this can be accomplished by introducing a random Pauli cycle $\mathcal{R}$ at each time step between each application of the cycle of interest $\mathcal{G}$[25,26]. This additional random Pauli cycle $\mathcal{R}$ comes with an additional overhead that will increase the number required gates to implement a given algorithm. RC has been developed to eliminate this overhead[21]. The resulting noise channel when using RC is then associated with the composition of $\mathcal{G}$ with a random Pauli cycle $\mathcal{R}$, called a dressed cycle $\mathcal{G}\mathcal{R}$, which is an important characterization primitive for any algorithm implemented via RC[21]. Therefore CB estimates the average of the process fidelities of the dressed cycle $\tilde{\mathcal{G}}\tilde{\mathcal{R}}$

$$F_{\mathrm{RC}}(\tilde{\mathcal{G}}, \mathcal{G}) = \sum_{\mathcal{R} \in \{\mathcal{I}, \mathcal{X}, \mathcal{Y}, \mathcal{Z}\}^{\otimes N}} 4^{-N} F(\tilde{\mathcal{G}}\tilde{\mathcal{R}}, \mathcal{G}\mathcal{R}). \quad (4)$$

In addition to the dressed cycle fidelity, the process fidelity of the noisy gate $\tilde{\mathcal{G}}$ alone is of interest. The process fidelity of a specific gate $\tilde{\mathcal{G}}$ may be estimated by taking the ratio of the estimates obtained for $\tilde{\mathcal{G}}$ and the identity process $\tilde{\mathcal{I}}$, in analogy to interleaved benchmarking[27]. It should be noted that this method of estimating the fidelity of the noise on $\tilde{\mathcal{G}}$ alone is generally subject to a large systematic uncertainty[28], so the CB method is most precise in the important context of characterizing errors on dressed cycles[21].

CB can be used to efficiently characterize non-Clifford gates by selecting random gates and correction operators using RC[21]. However, the general protocol for non-Clifford gates is more complex, so a simplified version for characterizing the errors occurring under a fixed cycle of Clifford gates $\mathcal{G}$ composed with a random Pauli cycle $\mathcal{R}$ is as follows (the protocol is illustrated in Fig. 1, where we explain the motivation for each step further below):

1. Select a set of $N$-qubit Pauli matrices $\mathsf{P}$ with $K = |\mathsf{P}|$ elements.
2. Select two lengths $m_1$ and $m_2$ such that the multiple application of $\mathcal{G}$ composes to the identity $\mathcal{G}^{m_1} = \mathcal{G}^{m_2} = \mathcal{I}$.
3. Perform the following sequence for each Pauli matrix $P \in \mathsf{P}$, length $m \in (m_1, m_2)$, and $l \in (1, \dots, L)$, where $L$ describes the number of random sequences per Pauli.

3a. Select $m + 1$ random $N$-qubit Pauli cycles $\mathcal{R}_0, \mathcal{R}_1, \dots, \mathcal{R}_m$, and define the randomized circuit

$$\mathcal{C}(P) = \mathcal{R}_m \mathcal{G} \mathcal{R}_{m-1} \mathcal{G} \dots \mathcal{R}_1 \mathcal{G} \mathcal{R}_0 \quad (5)$$

as illustrated in Fig. 1.

3b. Calculate the expected outcome of the sequence $\mathcal{C}(P)$ assuming ideal gate implementations.

3c. Main experiment: Implement $\mathcal{C}(P)$ and estimate the overlap

$$f_{P,m,l} = \mathrm{Tr}[\mathcal{C}(P) \, \tilde{\mathcal{C}}(\rho)] \quad (6)$$

between the expected outcome and the noisy implementation $\tilde{\mathcal{C}}(\rho)$ for some initial state $\rho$ that is a $+1$-eigenstate of $P$. State preparation and measurement are realized by applying the operations $\tilde{\mathcal{B}}_P$ and $\tilde{\mathcal{B}}_{\mathcal{C}(P)}^{\dagger}$ that are described in Supplementary Note 2.

4. Estimate the composite process fidelity via

$$F_{\mathrm{RC}}(\tilde{\mathcal{G}}, \mathcal{G}) = \sum_{P \in \mathsf{P}} \frac{1}{|\mathsf{P}|} \left( \frac{\sum_{l=1}^{L} f_{P,m_2,l}}{\sum_{l=1}^{L} f_{P,m_1,l}} \right)^{\frac{1}{m_2 - m_1}}. \quad (7)$$

Step 1 ensures that the action of the $N$-qubit process is accurately estimated. In Supplementary Note 5, we prove that the uncertainty of the fidelity estimate is independent of the number of qubits $N$, and the number of Pauli matrices $K$ that need to be sampled depends only on the desired precision. This highlights the scalability of the protocol for large quantum processors.

Step 2 ensures that the measurement procedures for circuits in Eq. (8) with two different values of $m$ are the same. Having the same measurement procedures for the two values of $m$ is crucial to decouple the SPAM errors from the decay in the process fidelity via the ratio in Eq. (7). In our experiment, we always choose $m_1 = 4$ and $m_2$ to be an integer multiple of 4, as, for the considered gates, applying the operation four times subsequently yields the identity process $\mathcal{G}^4 = \mathcal{I}$.

In step 3a, we choose random Pauli cycles to engineer an effective Pauli noise process across the $L$ randomizations. This enables us to extract a process fidelity from the decay of $\sum_{l=1}^{L} f_{P,m,l}/L$ with the sequence length $m$. Note that unlike typical

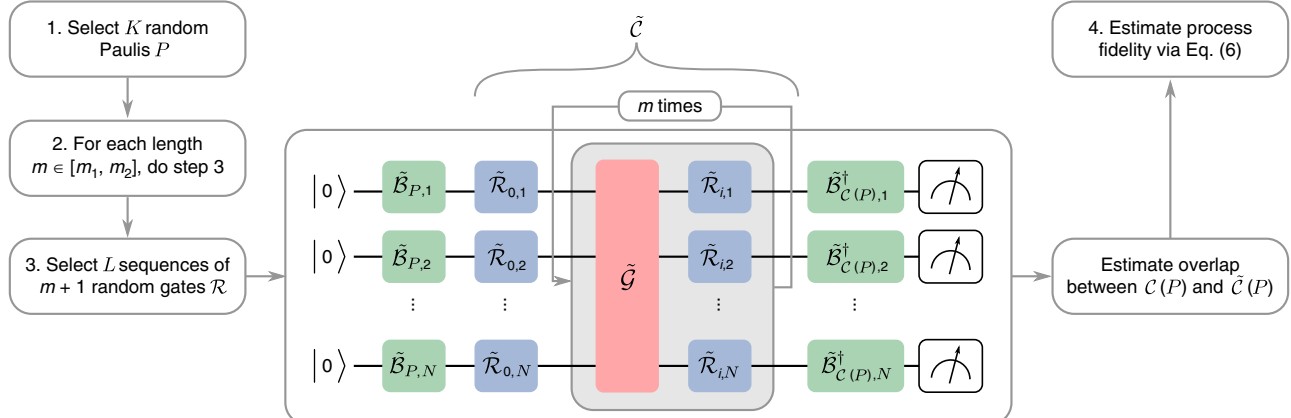

**Fig. 1** Schematic circuit implementation of the experimental cycle benchmarking protocol. The protocol can be subdivided into three parts, depicted by the different colors. The green gates $\tilde{\mathcal{B}}$ describe basis changing operations for the state preparation and the measurement (SPAM) procedure. The red gates $\tilde{\mathcal{G}}$ are the noisy implementations of some gate of interest (in this work, the global Mølmer–Sørensen gate acting on all qubits). The blue gates are random Pauli cycles that are introduced to create an effective Pauli channel per application of the gate of interest, where $\tilde{\mathcal{R}}_{i,j}$ denotes the $j$th tensor factor of the $i$th gate. Creating an effective Pauli channel per application enables errors to be systematically amplified under $m$-fold iterations for more precise and SPAM-free estimation of the errors in the interleaved red gates $\tilde{\mathcal{G}}$. The blue and the red gates together form the random circuit $\tilde{\mathcal{C}}$. The sequence of local operations before the first and last rounds of random Pauli cycles are identified as conceptually distinct but were compiled into the initial and final round of local gates in the experiment. The experimental parameters $K, m$, and $L$ of this work and the exact definitions of $\tilde{\mathcal{B}}$ and $\tilde{\mathcal{R}}$ are given in Supplementary Note 7.

randomized benchmarking protocols, the above protocol does not have an inversion gate. Formally, the final random Pauli can be regarded as a correction gate for the random Pauli gates in the rest of the circuit composed with another random Pauli that we use to isolate exponential decays as in character benchmarking[29].

In step 3b, for any Clifford cycle $\mathcal{G}$, Pauli matrix $P$, and Pauli cycles $\mathcal{R}_0, \ldots, \mathcal{R}_m$, the expected outcome of the ideal implementation $\mathcal{C}(P)$ is a Pauli matrix that can be efficiently calculated. Note that only the sign of $\mathcal{C}(P)$ depends on the random Pauli cycles. This sign is accounted for when estimating the expectation value with the procedure outlined in Supplementary Note 2. Incorporating the sign engineers a measurement of the expectation value of $\mathcal{C}(P)$ that is robust to SPAM errors, as otherwise the expectation values result from a multi-exponential decay[24,29].

In step 3c, we experimentally prepare an eigenstate of a Pauli matrix $P$, apply a circuit $\tilde{\mathcal{C}}$ with interleaved random Pauli cycles, and measure the expectation value of $\mathcal{C}(P)$. The explicit procedures we use for preparing the eigenstate and measuring the expectation value are described in Supplementary Note 2. As discussed in Supplementary Note 5, the number of measurements required to estimate the expectation value to a fixed additive precision is independent of the number of qubits.

As we prove in Supplementary Note 4, the expected value of $F_{RC}(\tilde{\mathcal{G}}, \mathcal{G})$ in Eq. (7) for two values of $m_1$ and $m_2$ as in step 2 is equal to the composite process fidelity $F_{RC}(\tilde{\mathcal{G}}, \mathcal{G})$ in Eq. (4) up to $\mathcal{O}\left(\left[1 - F_{RC}(\tilde{\mathcal{G}}, \mathcal{G})\right]^2\right)$ and always provides a lower bound.

**Experimental results**. We demonstrate the practicality of CB for multi-qubit systems by using it to experimentally estimate the process fidelity of cycles acting globally on quantum registers containing 2, 4, 6, 8, and 10 qubits. The specific cycles we consider consist of simultaneous local Pauli gates and multi-qubit entangling MS gates[22,23] combined with simultaneous local Pauli gates. We confine $^{40}\text{Ca}^+$ ions in a linear Paul-trap and encode a single qubit in the electronic states of each atomic ion. The encoding utilizes the $|0\rangle = 4S_{1/2}(m_j = -1/2)$ ground-state and the $|1\rangle = 3D_{5/2}(m_j = -1/2)$ metastable excited state. Our quantum computing toolbox comprises independent arbitrary single-qubit operations and fully entangling $N$-qubit MS gates, acting on all $N$ qubits in the register simultaneously (see Supplementary Note 7). An experimental run consists of: (i) Doppler cooling, (ii) sideband-cooling of the two motional modes with lowest frequencies, (iii) optical pumping to the initial state $|0\rangle^{\otimes N}$, (iv) coherent manipulation, and (v) readout of the ions. Each sequence is repeated 100 times to gather statistics (for experimental details, see Supplementary Note 7 and ref. [30]).

Under Markovian noise, the estimate of the process fidelity from Eq. (7) is independent of the sequence lengths $m_1$ and $m_2$ used to estimate it (see Supplementary Note 3). We tested whether our experimental apparatus satisfied this assumption by performing measurements at three values of $m$ (4, 8, and 12) on a register containing 6 qubits and comparing the results obtained from pairs of sequence lengths against each other. The data are tabulated in Supplementary Table 2, where the variation of the estimated fidelities is within 0.1%, which is smaller than the corresponding uncertainties of 0.4%. This suggests that the errors are Markovian and the estimated process fidelity is independent of the chosen sequence lengths for our system and henceforth we only use two sequence lengths to estimate the process fidelity.

The CB protocol is practical to implement on large processors because the fidelity can be accurately estimated using a number of Pauli matrices that is independent of the number of qubits $N$ (see Supplementary Note 5). To illustrate the rapid convergence under

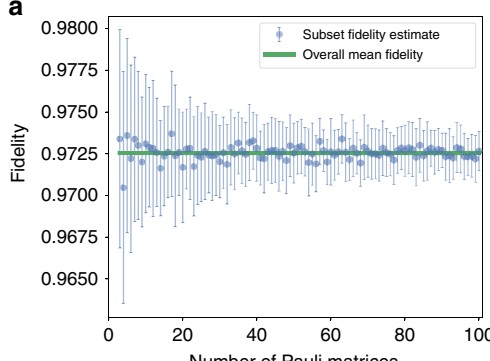

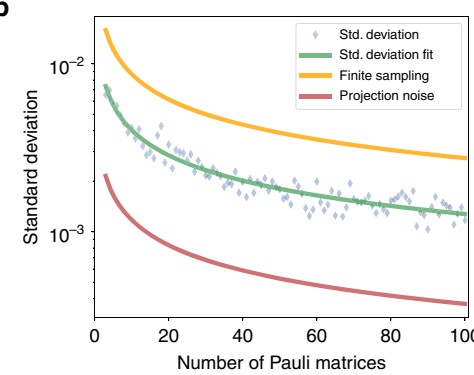

**Fig. 2** Experimental evidence demonstrating rapid convergence under finite sample size with favorable constant factors. **a** Mean fidelity estimates from 30 randomly sampled subsets of Pauli matrices as a function of the size $K$ of the subset. The error bars illustrate the standard deviation of the 30 samples, that is, the standard error of the mean. The green line describes the mean fidelity $F = 97.25(8)\%$ calculated from the complete data set. **b** The standard deviation of the fidelity from plot **a** against $K$ including a bound due to finite sampling of Pauli channels $\sigma_{\text{Pauli}} = 0.0275(8)/\sqrt{K}$ in orange, a fit of the standard deviation $\sigma = 0.0127(2)/\sqrt{K}$ in green, and a fit of the expected projection noise $\sigma_{\text{lower}} = 0.00375(1)\sqrt{K}$ in red (see Supplementary Note 5).

finite sample size, we performed CB of local Pauli operations on a 4-qubit register by exhaustively estimating all $4^4 - 1 = 255$ possible decay rates. We estimate the average fidelities via Eq. (7) for multiple subsets $\mathsf{P}$ of the set of all Pauli matrices. For each $K = 1, \ldots, 100$, we evaluate the fidelity for 30 randomly chosen subsets $\mathsf{P}$ containing $|\mathsf{P}| = K$ Pauli matrices. The mean and standard deviation of the estimated fidelities as functions of the subset size are shown in Fig. 2. In Fig. 2b, we introduce two boundaries between which the observed standard deviation should lie if we are choosing appropriate sequence lengths and sample sufficiently many random circuits per sequence length. For the lower bound, we assume quantum projection noise to be the only noise source. We evaluate the shot noise for the measured data and perform error propagation to calculate the lower bound $\sigma_{\text{lower}} = 0.00375(1)/\sqrt{K}$. This lower limit could be reached if the noise in the system is completely isotropic (e.g., global depolarizing). Biased noise or drift (see Supplementary Note 10) will lead to uncertainties bigger than those originating from quantum projection noise. We furthermore test that the fluctuations between different Pauli channels is bounded by an error model that assumes worst-case fluctuations between channels. This bound does not depend on the register size but only on the fidelity $F$ and can be estimated via $\sigma_{\text{Pauli}} = (1 - F)/\sqrt{K} = 0.0275(8)/\sqrt{K}$ (see Supplementary Note 5).

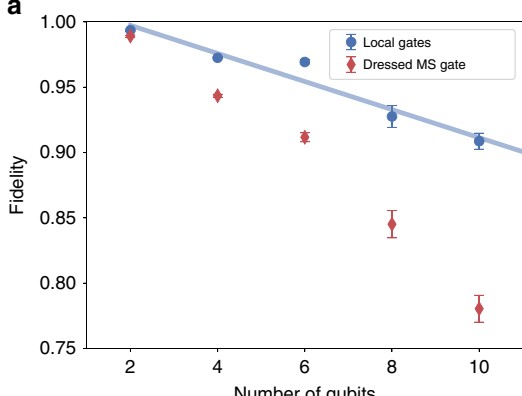

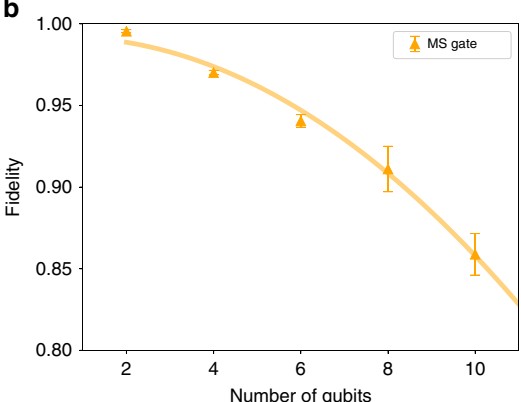

**Fig. 3** Experimental estimates of how rapidly error rates increase as the processor size increases. **a** Process fidelities obtained under cycle benchmarking for local gates (blue circles) and for sequences containing dressed Mølmer–Sørensen (MS) gates (red diamonds), that is, MS gates composed with a random Pauli cycle, plotted against the number of qubits in the register. The local operations are consistent with independent errors fitted according to Eq. (8). **b** Estimate of the process fidelity of an MS gate obtained by taking the ratio of dressed MS and local process fidelities. The data are fitted to Eq. (9) and is consistent with a constant error per two-qubit coupling.

**Table 1 Process fidelities (%) estimated via CB. Measured fidelities for local gates, dressed MS gates, and the inferred MS gate fidelity as depicted in Fig. 3.**

| Qubits | Local gates | Dressed MS gate | MS gate |
|---|---|---|---|
| 2 | 99.37 (7) | 98.92 (8) | 99.6 (1) |
| 4 | 97.25 (8) | 94.3 (1) | 97.0 (2) |
| 6 | 96.9 (2) | 91.2 (3) | 94.1 (4) |
| 8 | 92.8 (8) | 85 (1) | 91 (2) |
| 10 | 90.9 (6) | 78 (1) | 86 (2) |

The observed standard error of the mean $\sigma = 0.0127(2)/\sqrt{K}$ is larger than the lower bound given by quantum projection noise but smaller than the worst-case bound from sampling finite Pauli channels. The data demonstrate that we can estimate the process fidelity $F$ to an uncertainty smaller than $(1 - F)/\sqrt{K}$ independent of the register size with other experimental parameters held fixed (the parameters are listed in Supplementary Table 1).

We performed CB on local operations and with an interleaved MS gate on registers containing 2, 4, 6, 8, and 10 qubits. The process fidelity as a function of the number of qubits in the register is shown in Fig. 3 and Table 1. While it is expected that the fidelity over the full register decreases with increasing register size, an important question is whether the effective error rate per qubit increases or significant cross-talk effects appear, with increasing numbers of qubits.

We observe that the fidelity for local CB (blue circles in Fig. 3a) decays linearly with register size $N$, as

$$F = 1 - \epsilon_P N, \tag{8}$$

with $\epsilon_P = 0.011(2)$. The linear decay of the fidelity indicates that our single-qubit Pauli operations do not show increasing error rates per qubit or a significant onset of cross-talk errors as the register size increases. Each single-qubit Pauli operation requires $n_S$ native gates, where on average $\langle n_S \rangle = 1.27$, independent of the system size. Therefore, the effective process fidelity of a native single-qubit gate is $1 - \epsilon_P/\langle n_S \rangle = 0.992(1)$.

The CB measurements with interleaved MS gates give the process fidelity of the MS gate composed with a round of local randomizing gates as in Eq. (4) (a dressed MS gate, see red diamonds in Fig. 3a). This determines the error rate when a circuit is implemented by RC[21]. The process fidelity of the interleaved gate can be estimated by the ratio of the dressed MS and local fidelities as in interleaved randomized benchmarking[27]. The resulting estimates are plotted in Fig. 3b. We note that these estimates may have a large systematic error that is on the same order as the overall error rate[28]. This systematic uncertainty primarily arises due to coherent over- and under-rotations with similar rotation axes. The MS gate performs rotations around the non-local axes $\sigma_x^{(i)} \otimes \sigma_x^{(j)}$, which are substantially different from the single-qubit rotation axes. Therefore, it is unlikely that any coherent errors on the MS gate accumulate with the errors on the single-qubit rotations, and so we neglect this systematic error. We conjecture that the process fidelity of the MS gate should decay quadratically due to an error in each of the $\binom{N}{2}$ couplings between pairs of qubits introduced by the MS gate. If we assume an average error rate $\epsilon_2$ per two-qubit coupling, we can describe the MS gate fidelity as

$$F_{MS} = 1 - \epsilon_2 \frac{N^2 - N}{2}. \tag{9}$$

Fitting this model to the results in Fig. 3b gives an estimated error per two-qubit coupling of $\epsilon_2 = 0.0030(2)$. However, we cannot harness these two-qubit couplings individually in the experiment and thus they cannot be compared to individually available gates. The deviations of the fidelity estimates from the model defined in Eq. (9) are within the expected statistical uncertainty and we believe that these deviations arise mainly from day-to-day fluctuations in the experiment.

## Discussion

In summary, we have developed CB and demonstrated its practicality by implementing it on quantum registers containing $N = 2$, 4, 6, 8, and 10 qubits. In comparison, a single random Clifford gate for 8 and 10 qubits would require >50 MS gates and so randomized benchmarking for 8 and 10 qubits would require a large number of measurements to achieve a useful statistical precision. CB is practical in regimes where randomized benchmarking is impractical because it uses local randomizing gates. A similar approach was independently considered in refs. [29,31] to characterize a two-qubit Clifford gate. However, the approach implemented here and proposed previously in ref. [26] can be applied in a scalable manner to processors with arbitrary numbers of qubits.

The total experimental time and post-processing resources required for our implementation were approximately independent of the number of qubits (see Supplementary Table 1), after

accounting for the additional tests performed on specific numbers of qubits. This is achieved because, as we provide proof in Supplementary Note 5, the uncertainty of the fidelity estimate is independent of the number of qubits $N$, and the number of Pauli matrices $K$ that need to be sampled depends only on the desired precision. In addition, we demonstrated experimentally that the estimate of the fidelity and its error converges rapidly under finite sample size (Fig. 2) and that the estimated fidelities are approximately independent of the sequence lengths used. The data from CB also gives estimates of the diagonal of the Pauli–Liouville representation of the effective noise. A natural open question is to use this procedure to reconstruct the underlying noise model, which we leave for future work.

CB can be readily implemented on general quantum computing architectures to estimate the fidelity of multi-qubit processes. The fidelity corresponds to the effective error rate under RC[32]. It should be noted that the performance of the same operation in a circuit without RC can differ significantly from the estimated fidelity of its constituents due to the addition or cancellation of coherent errors[33]. This is a general issue with performance metrics for quantum operations[34] and we want to emphasize that RC has been designed to eliminate these coherent errors. The protocol also provides insight into how noise scales within a fixed architecture. In our ion trap, the fidelity of local gates across the whole register decreased linearly with $N$, demonstrating that our native single-qubit gates have an average fidelity of 99.2(1)% and do not deteriorate with the register size. Thus we have demonstrated a scalable method to validate a major requirement for fault-tolerant quantum computation. In addition, we performed interleaved CB protocols to estimate the performance of the multi-qubit entangling MS gate. From the ratio between the dressed MS and the local CB fidelities, we infer entangling gate fidelities ranging from 99.6(1)% to 86(2)% for 2–10 qubits. While this inference is in principle subject to a large systematic uncertainty[24], we have argued that the systematic uncertainty should be small for our set of operations. We leave the problem of quantifying or reducing this systematic uncertainty open, but note that a natural approach would be to quantify how coherent the errors are using generalizations of either purity benchmarking[35] or iterated interleaved benchmarking[36].

## Data availability
The data that support the findings of this study are available from the corresponding author upon reasonable request.

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

## Acknowledgements
We gratefully acknowledge support by the Austrian Science Fund (FWF), through the SFB Fo-QuS (FWF Project No. F4002-N16), as well as the Institut für Quanteninformation GmbH. In addition, we acknowledge support from the Austrian Research Promotion Agency (FFG) contract 872766. This research was funded by the Office of the Director of National Intelligence (ODNI) and Intelligence Advanced Research Projects Activity (IARPA) through the Army Research Office grant W911NF-16-1-0070. All statements of fact, opinions, or conclusions contained herein are those of the authors and should not be construed as representing the official views or policies of IARPA, the ODNI, or the U.S. Government. We also acknowledge support by U.S. A.R.O. through grant W911NF-14-1-0103. This research was undertaken thanks in part to funding from TQT, CIFAR, the Government of Ontario, and the Government of Canada through CFREF, NSERC, and Industry Canada. We want to thank the anonymous referees for their valuable remarks.

## Author contributions

A.E., J.J.W., P.S., T.M., J.E. and R.B. wrote the manuscript and provided revisions. J.J.W., T.M. and P.S. developed the research based on discussions with J.E. and R.B. J.J.W. and J.E. developed the theory. A.E., E.M. and P.S. performed the experiments. A.E., E.A.M., P.S., L.P., M.M. and R.S. contributed to the experimental set-up. A.E. and J.J.W. analyzed the data. All authors contributed to discussions of the results and the manuscript.

## Competing interests

J.J.W. and J.E. are founding members of Quantum Benchmark Inc. T.M. and R.B. are founding members of Alpine Quantum Technologies GmbH.
