## [Peer Review File · Nature Communications]

Reviewers' comments:

Reviewer #1 (Remarks to the Author):

The manuscript "Characterizing large-scale quantum computers via cycle benchmarking" by A. Erhard et. al. presents a new technique, called cycle benchmarking (CB), for characterizing quantum computers, and demonstrates it experimentally on an ion trap quantum computer. CB represents a new and useful contribution to the quantum characterization and diagnostic toolkit, alongside existing techniques such as full tomography, direct fidelity estimation, randomized benchmarking (RB), gate set tomography (GST), and others. Unlike previous techniques, it combines favorable scaling to larger numbers of qubits with systematic process fidelity estimation that is independent of state preparation and measurement (SPAM) errors. As qubit numbers grow, and algorithms are attempted whose outcomes are unknown because they cannot be classically simulated, the ability to characterize the fidelities of component quantum operations with SPAM-free techniques that are practical for large qubit numbers - as CB does - will be crucial.

I have several points of clarification and additional discussion the authors should address. Once addressed, this manuscript merits publication in Nature Communications.

*Some discussion of the assumptions required for CB would be appropriate to include in the main text, perhaps with additional discussion in the supplementary material. For example, RB assumes all errors on Clifford gates are the same, which is a problematic assumption for many realistic experimental setups. Some discussion of a similar but weaker assumption needed for CB is in the supplementary material, but this should be fleshed out further (how valid is this weaker assumption for realistic experiments? What could mitigate impacts of this assumption?), and a brief description of this and any other assumptions made should be included in the main text.

*How many experimental trials per CB sequence are necessary to achieve a given uncertainty on the outcome of the fidelity analysis? While I see "Each sequence is repeated 100 times to gather statistics", it is not explained why this value is chosen. How does this scale with the number of qubits, or with the precision desired? Other techniques (tomography, RB) require many sequence or measurement repetitions to achieve high precision on reported fidelities, and more trials are required to achieve lower uncertainties, so this is an important point of comparison.

*"In the Supplementary Information we prove that the number of Pauli matrices that need to be sampled is independent of the number of qubits, highlighting the scalability of the protocol for large quantum processors." It was unclear where this proof was in the supplementary information. If erroneously omitted, this proof should certainly be included. It should additionally be clearly labeled and easy to find, as it provides crucial support to the claim that CB is scalable to large numbers of qubits.

*Are the MS gates characterized with CB experimentally performed as a global MS gate with all 2, 4, 6, 8, or 10 qubits, or are all but 2 qubits shelved in some way to perform an MS gate on the unshelved 2 qubits?

*Figure 1: the color coordination between the outer step-by-step text and the gates in the circuit make it look like steps 1 and 4 have a relation to the green-colored gates, and steps 2, 3, unlabeled have a relation to the blue-colored gates; it is not clear that this is actually the case and should be clarified. Separately, the text in box 2 ends with a transitive verb that requires an object, eg "For each length m in $[m_1, m_2]$, do Step 3." A colon after "do" would also work.

*Figure 2: the plots should have legends, in addition to the explanations in the caption.

*"[8] The process fidelity, also known as the process fidelity..." The redundancy in this footnote should

be fixed.

Reviewer #2 (Remarks to the Author):

The authors developed a variant on the RBM technique, called cycle benchmarking (CB). The author show that, in contrast to RBM, the number of Pauli operators one needs to sample in the CB is independent of the number of qubits. This feature makes this protocol very appealing and potentially can be used in practice, as claimed by the authors, to extract gate infidelities (in a SPAM-robust way) for large-scale quantum processors. The author present the CB protocol in the main text and provide a rigorous derivations in the appendix. The protocol is presented very clearly (to be followed by others who are interested in implementing it) and the figure certainly helps. I went carefully through the derivations in subsections B and C of the appendix, and read through the rest.

The experimental part of the contribution is impressive, as it is the first application, to my knowledge, of a RBM-like technique to a system composed of more than a handful of qubits.

I believe this technique is will be interesting and useful to many labs developing quantum information processors, especially in this NISQ era.

I therefore recommend publication in Nature Communications.

Reviewer #3 (Remarks to the Author):

The manuscript NCOMMS-19-09040-T describes a method for characterizing the fidelity of multi-qubit operations in the context of quantum computation and an experimental demonstration in a trapped-ion system. The research presented is novel and highly topical. The characterization of quantum operations is a subject of great importance as system sizes grow; existing approaches have considerable weaknesses when applied to many qubits. The experimental results are convincing.

I do, however, find the manuscript very difficult to read. I have read it with the background of an experimentalist familiar with the general concepts of tomography and randomized benchmarking. In particular the difficulty to parse the first part of the manuscript and relate it to the state of the art has resulted in a considerable delay in the submission of my report. I think in the end I got the idea of how the scheme works, but I believe the presentation must be considerably improved.

While the introduction on the first page is quite understandable, the discussion on the second and third page should be made more accessible in my opinion, in particular the paragraph containing the motivation for and procedure related to the effective Pauli noise channel introduced. Methods such as randomized compiling etc should be briefly explained when they are referred to. The implementation section of the proposal is again quite clear.

A significantly revised version of the manuscript along these lines may be suitable for NCOMMS; otherwise, I would recommend a journal that focuses on QIP.

Reviewer #4 (Remarks to the Author):

This work describes a method to characterize the performance of a quantum logic gate in a realistic quantum computer consisting of a (moderate-to-)large number of qubits. This method is applied to an ion trap quantum processor with up to 10 qubits. The full characterization via the gate process tomography is known to be impractical (or, intractable) as the system size scales, so a practically useful method for characterizing the operation of quantum computers at larger scale is of great

interest to the research community. From that standpoint, this paper addresses an important topic in a timely manner. I have some topics that the authors can help clarify before the manuscript is ready for publication in Nature Communications.

1. Some clarifications on the efficacy of the overall method: On Page 1, 4th paragraph, the authors say that cycle benchmarking (CB) is "a protocol for estimating the effect of all global and local error mechanisms" that occur when clock cycle of operations is applied to a quantum register. What is (are) the "effect(s)" that it estimates? Is it limited to a single number (process fidelity) for each global and local gate errors, or can it say more about the nature of the error mechanisms? If it can say more, what are the types of error mechanisms and their effects can this method provide?

2. Along similar lines, the description of the protocol in Page 2 and Figure 1 shows the CB process for a Clifford gate. However, the introductory description of the protocol starts without the constraints that the gate G it applies to must be in the Clifford group. What is the general gate set over which the CB applies? Is CB more generic than the protocol outlined in Page 2 and Figure 1 (i.e., is this a restricted example), or does CB apply only to Clifford group gates?

3. In the experimental example describing Fig. 2b, the authors give two limits, namely the "lower bound given by quantum projection noise" and the "upper bound contributed from sampling a finite number of Pauli matrices". Where do these bounds come from, and experimentally, what provides this bound? I was looking for additional information in the Supplementary Information, but there were no further details there. For example, can one reduce the upper bound by increasing the number of Pauli matrices they sample? The question really is how fundamental these two bounds are, and whether there is any insights as to what constitutes this gap, and whether other experimental configurations can tighten these bounds.

4. At the bottom of the left column in Page 4 regarding Fig. 3b, the authors state that the estimates may have a large systematic error that is on the same order as the overall error rate, which arises from the over- and under-rotations of the gates. Is there any indicator from the analysis of the data by the CB protocol that leads them to this conclusion, and if so, what insight does the tool/method provide? If not, how do the authors arrive at this conclusion (both the source and the order-of-magnitude)?

5. More generally, does the CB method provide a way to distinguish the Markovian vs systematic noise (due to drifts in various parameters of the experiment)? Although the experiments were not done, it is reasonable to believe that the noise processes in ion qubits might be very different from other solid-state qubits. Would the CB method be useful in providing better understanding on the difference in the nature of the noise sources? (Actually, this question is not necessarily a criticism on the manuscript, but more from my curiosity. I would love to hear an answer, but understand if this discussion is outside the scope of this paper.)

6. Eq. (8) seems to apply really well to the cases when the number of qubits is 2, 8 and 10, but not so well for 4 and 6 qubit cases. Anything to take away from here?

7. I find some typos and inconsistencies in the paper and the supplementary material, which the authors should fix. Including but not limited to

a. End of Page 3, the sentence in "(ii) sideband-cooling..." seems broken.

b. In Supplementary material, in Section D after the first (un-numbered) equation, it says $\Delta m \sim 1-F(E,I)$, but given Δm is an integer presumably much larger than 1, it looks like the right hand side should be $1/(1-F(E,I))$.

c. In the same sentence, it says " $\sigma_P \propto \dots$ " The right hand side of that proportionality does not seem to make sense (undefined quantity?).

Questions from Reviewer 1

1. *Some discussion of the assumptions required for CB would be appropriate to include in the main text, perhaps with additional discussion in the supplementary material. For example, RB assumes all errors on Clifford gates are the same, which is a problematic assumption for many realistic experimental setups. Some discussion of a similar but weaker assumption needed for CB is in the supplementary material, but this should be fleshed out further (how valid is this weaker assumption for realistic experiments? What could mitigate impacts of this assumption?), and a brief description of this and any other assumptions made should be included in the main text.*

The only assumptions made in CB are that all noise is Markovian on the timescale of a cycle and that the noise on each cycle of independent single-qubit gates is independent of the specific gates being implemented. We include a revised statement of these assumptions in the main text (page 2, paragraph 2) and a clearer section on the mathematical assumptions in the supplementary information. We note that these are formally the same assumptions under which randomized benchmarking was originally analyzed, although more recent treatments (e.g., of Ref 33) have removed the gate-independent assumption for randomized benchmarking. However, as we state in the supplementary information, the gate-independent assumption is applied to a smaller set of gates, namely, to cycles of single-qubit gates rather than rounds with varying numbers of CNOT gates and so is a more realistic assumption. Additionally, we are aware of work by independent authors that the gate-independent assumption can also be removed using the techniques of Ref 33. However, we are not currently able to review and cite the proof and so make a weaker statement that we expect the assumption can be removed.

2. *How many experimental trials per CB sequence are necessary to achieve a given uncertainty on the outcome of the fidelity analysis? While I see “Each sequence is repeated 100 times to gather statistics”, it is not explained why this value is chosen. How does this scale with the number of qubits, or with the precision desired? Other techniques (tomography, RB) require many sequence or measurement repetitions to achieve high precision on reported fidelities, and more trials are required to achieve lower uncertainties, so this is an important point of comparison.*

Interestingly, the uncertainty on the achieved average fidelity does not scale with the number of qubits. The 100 experimental shots are chosen such that the delay of the experiment when changing experimental sequences does not dominate the measurement time. We added a section „Finite sampling effects“ to the supplementary information that gives a more detailed treatment of the required number of experimental trials, specifically showing that the number of trials is independent of the number of qubits in contrast to tomography, which requires a number of trials that grows exponentially with the number of qubits.

3. *In the Supplementary Information we prove that the number of Pauli matrices that need to be sampled is independent of the number of qubits, highlighting the scalability of the protocol for large quantum processors." It was unclear where this proof was in the supplementary information. If erroneously omitted, this proof should certainly be included. It should additionally be clearly labeled and easy to find, as it provides crucial support to the claim that CB is scalable to large numbers of qubits.*

We have updated the supplementary material. The proof can now be found in the section „Finite sampling effects“. We furthermore added specific links to this chapter in the supplementary information. In addition, we included the section “Testing the dependence of the fidelity uncertainty on the register size” in the supplementary information, which shows that the uncertainties of our fidelity estimates are independent of the number of qubits.

4. *Are the MS gates characterized with CB experimentally performed as a global MS gate with all 2, 4, 6, 8, or 10 qubits, or are all but 2 qubits shelved in some way to perform an MS gate on the unshelved 2 qubits?*

We experimentally implement the MS gate as global N-Body interaction. We addressed this in the main text (page 4, paragraph 1 and in the caption of fig. 3) and we made the “Experimental methods” in the supplementary information more detailed.

5. *Figure 1: the color coordination between the outer step-by-step text and the gates in the circuit make it look like steps 1 and 4 have a relation to the green-colored gates, and steps 2, 3, unlabeled have a relation to the blue-colored gates; it is not clear that this is actually the case and should be clarified. Separately, the text in box 2 ends with a transitive verb that requires an object, eg “For each length m in $[m_1, m_2]$, do Step 3.” A colon after “do” would also work.*

The color-coding of the step-by-step description was indeed misleading. We removed the color coding in the revised manuscript.

6. *Figure 2: the plots should have legends, in addition to the explanations in the caption.*

We added legends to Figure 2 and updated the caption to make the relation clear.

7. *[8] The process fidelity, also known as the process fidelity...” The redundancy in this footnote should be fixed.*

We removed this footnote and put this information correctly into the main text (page 1, paragraph 2).

Questions from Reviewer 3

1. *I do, however, find the manuscript very difficult to read. I have read it with the background of an experimentalist familiar with the general concepts of tomography and randomized benchmarking. In particular the difficulty to parse the first part of the manuscript and relate it to the state of the art has resulted in a considerable delay in the submission of my report. I think in the end I got the idea of how the scheme works, but I believe the presentation must be considerably improved.*

While the introduction on the first page is quite understandable, the discussion on the second and third page should be made more accessible in my opinion, in particular the paragraph containing the motivation for and procedure related to the effective Pauli noise channel introduced. Methods such as randomized compiling etc should be briefly explained when they are referred to. The implementation section of the proposal is again quite clear.

In the introduction (page 1, paragraph 4) we reviewed randomized compiling in more detail to make the reader familiar with this concept. The mentioned discussion on page 2/3 is indeed quite technical/theoretical, which it needs to be in order to keep our statements valid and as precise as possible. However, we addressed the issue of not introducing certain concepts, by explaining the methods in the main text in more detail (page 2/3).

Questions from Reviewer 4

1. *Some clarifications on the efficacy of the overall method: On Page 1, 4th paragraph, the authors say that cycle benchmarking (CB) is “a protocol for estimating the effect of all global and local error mechanisms” that occur when clock cycle of operations is applied to a quantum register. What is (are) the “effect(s)” that it estimates? Is it limited to a single number (process fidelity) for each global and local gate errors, or can it say more about the nature of the error mechanisms? If it can say more, what are the types of error mechanisms and their effects can this method provide?*

In this work we focused on getting an estimate of one number, the process fidelity, as mentioned in the introduction (page 1, paragraph 2). As we added to the discussion section (page 6), based on the measured data we also get estimates of the diagonal of the Pauli-Liouville representation of the effective noise. With that information one could reconstruct the underlying noise model, which is outside the scope of this work.

2. *Along similar lines, the description of the protocol in Page 2 and Figure 1 shows the CB process for a Clifford gate. However, the introductory description of the protocol starts without the constraints that the gate G it applies to must be in the Clifford group. What is the general gate set over which the CB applies? Is CB more generic than the protocol outlined in Page 2 and Figure 1 (i.e., is this a restricted example), or does CB apply only to Clifford group gates?*

As correctly observed, the CB protocol itself is not limited to gates from the Clifford group. We chose to use only gates from Clifford group, because the presentation of the protocol becomes significantly simpler. We address this now in the main text (page 3, paragraph 2).

3. *In the experimental example describing Fig. 2b, the authors give two limits, namely the “lower bound given by quantum projection noise” and the “upper bound contributed from sampling a finite number of Pauli matrices”. Where do these bounds come from, and experimentally, what provides this bound? I was looking for additional information in the Supplementary Information, but there were no further details there. For example, can one reduce the upper bound by increasing the number of Pauli matrices they sample? The question really is how fundamental these two bounds are, and whether there is any insights as to what constitutes this gap, and whether other experimental configurations can tighten these bounds.*

In response to this and other related questions we updated the main text (page 4, last paragraph) and the section „Finite sampling effects“ in the supplementary information. The lower limit is a guide for the reader to know how small the uncertainty on the estimate of the process fidelity would be if the only error source was quantum projection noise. This limit can be lowered by increasing the number of repetitions R , but it can only be reached with isotropic (e.g. global depolarizing) noise. We also calculate a bound due to the finite sampling of different Pauli channels which scales with $1-F/\sqrt{K}$. This bound shows that the actual uncertainty depends on the fidelity F of the system as well as the number of subspaces K . One important aspect of this bound is its independence on the number of qubits N .

4. *At the bottom of the left column in Page 4 regarding Fig. 3b, the authors state that the estimates may have a large systematic error that is on the same order as the overall error rate, which arises from the over- and under-rotations of the gates. Is there any indicator from the analysis of the data by the CB protocol that leads them to this conclusion, and if so, what insight does the tool/method provide? If not, how do the authors arrive at this conclusion (both the source and the order-of-magnitude)?*

Estimating the fidelity of an interleaved gate may have large systematic errors, if the gates used for randomization and the interleaved gate have over- and under-rotations around similar rotation axis. But since the MS gate has quite a different rotation axis as the local gates used for randomization, we believe that the systematic error on the estimate on the MS gate fidelity is smaller than the overall error rate we observe, as mentioned in the main text (page 5). However, we do not have any data to rigorously support this belief at this stage. One could implement measurements to estimate the

systematic uncertainty of the interleaved gate (e.g. by iterative interleaved benchmarking), which is part of follow-up research, as added at the end of the paper (page 6-7).

5. *More generally, does the CB method provide a way to distinguish the Markovian vs systematic noise (due to drifts in various parameters of the experiment)? Although the experiments were not done, it is reasonable to believe that the noise processes in ion qubits might be very different from other solid-state qubits. Would the CB method be useful in providing better understanding on the difference in the nature of the noise sources? (Actually, this question is not necessarily a criticism on the manuscript, but more from my curiosity. I would love to hear an answer, but understand if this discussion is outside the scope of this paper.)*

Methods like a one-side test could flag the presence of non-Markovianity by, for example, a failure to fit a single exponential decay. But there exist no conclusive tests yet to properly detect non-Markovianity without assuming a specific model due to the vast number of possible models.

6. *Eq. (8) seems to apply really well to the cases when the number of qubits is 2, 8 and 10, but not so well for 4 and 6 qubit cases. Anything to take away from here?*

Eq. (9) represents a very simple error model for the MS gate. If one takes into account that the MS gate acting on N qubits, introduces $(N^2-N)/2$ 2-qubit couplings and furthermore one assumes that each 2-qubit coupling has a similar error rate, than one can describe the quadratic decay of the MS gate fidelity with the qubit number N . This gives only a qualitative picture of the behavior of the MS gate depending on the register size N and one could e.g. predict the fidelity of the MS gate if the register size would be larger than 10. In reality the fidelity of the MS gate depends on various experimental parameters, e.g. the Lamb-Dicke parameter, the trapping frequency, the number of motional modes which are present, the Gaussian laser beam coupling to the ions at the edge of the ion crystal. These parameters change with different ion numbers. A sophisticated error model including all experimental errors would be highly valuable, but is outside the scope of this paper. However, we see that the deviations from this simple model are within the expected statistical uncertainty and we believe that these deviations arise mainly from day-to-day fluctuations in the experiment, as added in the main text (page 6, paragraph 1).

7. *I find some typos and inconsistencies in the paper and the supplementary material, which the authors should fix. Including but not limited to*
 1. *End of Page 3, the sentence in "(ii) sideband-cooling..." seems broken.*
 2. *In Supplementary material, in Section D after the first (un-numbered) equation, it says $\Delta m \sim 1-F(E,I)$, but given Δm is an integer presumably much larger than 1, it looks like the right hand side should be $1/(1-F(E,I))$.*
 3. *In the same sentence, it says " $\sigma_P \propto \dots$ " The right hand side of that proportionality does not seem to make sense (undefined quantity?).*

We thank the referee for their attention to detail and have fixed these and other typos.

We fixed all the typos and inconsistencies given above and also we tried to fix as many irregularities as possible while working on the feedback.

Reviewers' comments:

Reviewer #1 (Remarks to the Author):

I am generally satisfied with the authors' clarifications and corrections in response to my comments. The author's alterations to the manuscript in response to comments from all of the reviewers have substantially improved the clarity and quality of the manuscript. In particular, the assumptions behind the CB technique and discussions of precision are now much more clear.

However, I am still a little unclear on point #3 I raised in my initial review. On page 3, the step 1 discussion claims, "In the Supplementary Information we prove that the number of Pauli matrices that need to be sampled is independent of the number of qubits, highlighting the scalability of the protocol for large quantum processors." While the proof and discussion in the Supplementary Information section "Finite sampling effects" has been much improved, to my reading, that section proves that the fidelity estimate uncertainty depends only on the actual fidelity, F , and the number of Pauli matrices sampled, K , and does not depend on the number of qubits, N . However, the quoted line in the main text appears to claim that the proof in that section shows that K does not depend on N , but the supplemental information section provides no discussion of K 's (lack of) dependence on N . It seems to me that K does not depend on N by construction, as the fidelity estimate uncertainty depends only on F and K , and therefore the value of K is a choice primarily driven by the level of precision desired, but I am not certain this interpretation is correct as the authors do not fully explicate this. The line in the main text is therefore inconsistent with the supplemental information, as I was straightforwardly expecting a proof that K does not depend on N . Therefore, either the line in the main text should be adjusted to reflect the actual content of the supplementary information section, or discussion should be added to the supplementary information section to clarify how the proof also shows that K does not depend on N .

Reviewer #4 (Remarks to the Author):

I am generally satisfied with the response by the authors and the revisions that are made accordingly, which addresses my questions from the first review.

The only question to which I do not see an adequate response is the estimation of the systematic error in the experiment (original question #4). The authors suggest that the rotation axis of MS gate is different from the local gates, and therefore the systematic errors (of over- and under-rotations) are cancelled via the randomization of the local gates. While all these make sense in their protocol, this also means that the actual fidelity of the MS gate, including the systematic errors, is lower than what is reported, as the gate error arising from the systematic errors are effectively cancelled out in the cycle benchmarking protocol. This type of cancellation might not be in full effect when a "real" circuit targeting an actual algorithm or application is implemented, i.e., the gate fidelity numbers reported by cycle benchmarking might not be available in real circuits. If that indeed is the case, there has to be some level of interpretation as to what the gate performance estimates (such as fidelity) this method provides means to the situation of a real experiment. I would appreciate an overall statement about this either in introduction or in conclusion.

Once this is addressed, I support publication of this manuscript in Nature Communications.

Remarks from Reviewer #1

1. *However, I am still a little unclear on point #3 I raised in my initial review. On page 3, the step 1 discussion claims, “In the Supplementary Information we prove that the number of Pauli matrices that need to be sampled is independent of the number of qubits, highlighting the scalability of the protocol for large quantum processors.” While the proof and discussion in the Supplementary Information section “Finite sampling effects” has been much improved, to my reading, that section proves that the fidelity estimate uncertainty depends only on the actual fidelity, F , and the number of Pauli matrices sampled, K , and does not depend on the number of qubits, N . However, the quoted line in the main text appears to claim that the proof in that section shows that K does not depend on N , but the supplemental information section provides no discussion of K 's (lack of) dependence on N . It seems to me that K does not depend on N by construction, as the fidelity estimate uncertainty depends only on F and K , and therefore the value of K is a choice primarily driven by the level of precision desired, but I am not certain this interpretation is correct as the authors do not fully explicate this. The line in the main text is therefore inconsistent with the supplementary information, as I was straightforwardly expecting a proof that K does not depend on N . Therefore, either the line in the main text should be adjusted to reflect the actual content of the supplementary information section, or discussion should be added to the supplementary information section to clarify how the proof also shows that K does not depend on N .*

We agree, that the quoted sentence from the main text could have been misleading and hence inconsistent with the Supplementary Information. We therefore adopted this sentence on page 3 and in the discussion on page 6 in the main manuscript.

In the Supplementary Information we prove that the uncertainty of the fidelity estimate is independent of the number of qubits N , and the number of Pauli matrices K that need to be sampled depends only on the desired precision. This highlights the scalability of the protocol for large quantum processors.

Remarks from Reviewer #4

1. *The only question to which I do not see an adequate response is the estimation of the systematic error in the experiment (original question #4). The authors suggest that the rotation axis of MS gate is different from the local gates, and therefore the systematic errors (of over- and under-rotations) are cancelled via the randomization of the local gates. While all these make sense in their protocol, this also means that the actual fidelity of the MS gate, including the systematic errors, is lower than*

what is reported, as the gate error arising from the systematic errors are effectively cancelled out in the cycle benchmarking protocol. This type of cancellation might not be in full effect when a "real" circuit targeting an actual algorithm or application is implemented, i.e., the gate fidelity numbers reported by cycle benchmarking might not be available in real circuits. If that indeed is the case, there has to be some level of interpretation as to what the gate performance estimates (such as fidelity) this method provides means to the situation of a real experiment. I would appreciate an overall statement about this either in introduction or in conclusion.

We agree with the referee and added a statement covering this issue in the discussion of the main manuscript on page 6.

It should be noted that the performance of the same operation in a circuit without RC can differ significantly from the estimated fidelity of its constituents due to the addition or cancellation of coherent errors. This is a general issue with performance metrics for quantum operations and we want to emphasize that RC has been designed to eliminate these coherent errors.